# Cytoprotective Activity of Newly Synthesized 3-(Arylmethylamino)-6-Methyl-4-Phenylpyridin-2(1*H*)-Ones Derivatives

**DOI:** 10.3390/molecules27175362

**Published:** 2022-08-23

**Authors:** Shynggys Sergazy, Zarina Shulgau, Aigerim Zhulikeyeva, Yerlan Ramankulov, Irina V. Palamarchuk, Ivan V. Kulakov

**Affiliations:** 1RSE “National Center for Biotechnology”, 13/5 Kurgalzhynskoe Road, Nur-Sultan 010000, Kazakhstan; 2National Laboratory Astana, Nazarbayev University, 53 Kabanbay Batyr Ave., Nur-Sultan 010000, Kazakhstan; 3Institute of Chemistry, Tyumen State University, 15a Perekopskaya St., 625003 Tyumen, Russia

**Keywords:** 3-aminopyridin-2(1*H*)-one derivatives, 3-(arylmethylamino)-6-methyl-4-phenylpyridin-2(1*H*)-ones, antiradical activity, cytoprotective activity

## Abstract

Currently, studies are being conducted on the possible role of the cytoprotective effect of biologically active substances in conditions of cerebral hypoxia or cardiomyopathies. At the same time, oxidative stress is considered one of the important mechanisms of cellular cytotoxicity and a target for the action of cytoprotectors. The aim of this study is to search for derivatives of 3-(arylmethylamino)-6-methyl-4-phenylpyridin-2(1*H*)-ones. The probability of cytoprotective action was assessed by measuring cell viability using two tests (with neutral red dye and MTT test). It was found that some derivatives of 3-(arylmethylamino)-6-methyl-4-phenylpyridin-2(1*H*)-ones under the conditions of our experiment had a pronounced cytoprotective activity, providing better cell survival in vitro, including the MTT test and conditions of blood hyperviscosity. To correlate the obtained results in vitro, molecular docking of the synthesized derivatives was also carried out. The standard drug omeprazole (co-crystallized with the enzyme) was used as a standard. It was shown that all synthesized derivatives of 3-(arylmethylamino)-6-methyl-4-phenylpyridin-2(1*H*)-ones had higher affinity for the selected protein than the standard gastro-cytoprotector omeprazole. The studied derivatives of 3-(arylmethylamino)-6-methyl-4-phenylpyridin-2(1*H*)-ones also fully satisfy Lipinski’s rule of five (RO5), which increases their chances for possible use as orally active drugs with good absorption ability and moderate lipophilicity. Thus, the results obtained make it possible to evaluate derivatives of 3-(arylmethylamino)-6-methyl-4-phenylpyridin-2(1*H*)-ones as having a relatively high cytoprotective potential.

## 1. Introduction

The search for new highly effective antioxidant and cytoprotective compounds is an urgent task for medical chemistry, since many diseases, such as diabetes mellitus, hepatitis of various etiologies, strokes, atherosclerosis, and many others occur against a background of pronounced oxidative stress and are accompanied by mass cell death. Previously, we have reported a method for the preparation of 4-aryl(hetaryl)-substituted 3-aminopyridin-2(1*H*)-ones based on the intramolecular cyclization of *N*-(3-oxoalkenyl)amides. Almost all of the obtained 3-aminopyridin-2(1*H*)-ones possess high antiradical activity [1]. In addition, 3-aminopyridin-2(1*H*)-one derivatives are of interest as potential biologically active compounds [2,3]. For example, amrinone is a pyridine phosphodiesterase 3 inhibitor [4]. Some 3-aminopyridin-2(1*H*)-one derivatives show antiviral activity, including against the AIDS virus [5,6]. The presence of an "embedded" amino acid fragment makes them attractive building blocks for the synthesis of novel derivatives with promising biological applications [7,8,9].

The primary amino group is one of the privileged reactive sites for potential modification of the obtained 4-aryl(hetaryl)-substituted 3-aminopyridine-2(1*H*)-ones. We have previously shown that the reaction of 3-amino-6-methyl-4-phenylpyridin-2(1*H*)-one **1** with aromatic aldehydes afforded the corresponding Schiff bases, the reduction of which with sodium borohydride led to the formation of 3-(arylmethylamino)-6-methyl-4-phenylpyridin-2(1*H*)-ones (Figure 1) [10].

Among a series of novel compounds, 3-(arylmethylamino)-6-methyl-4-phenylpyridin-2(1*H*)-one derivatives showed antiradical activity against DPPH and ABTS radicals. In addition, these synthesized compounds were tested in vivo for anxiolytic activity using a light–dark box test and antidepressant activity by Porsolt’s “behavioral despair” test. Several derivatives of 3-(arylmethylamino)-6-methyl-4-phenylpyridin-2(1*H*)-one with higher potential neurotropic activity than that of the comparator drugs (mexidol and amitriptyline) were found [11].

Undoubtedly, an important problem of pharmacology is the search for universal protection mechanisms that allow for protecting the cell from death in order to preserve its structural integrity and functional activity. It should be taken into account that one of the most common mechanisms of cell damage is the formation of reactive oxygen species that induce free radical lipid peroxidation of membranes with the development of oxidative stress [12,13]. In this regard, the detection of antiradical properties in newly synthesized substances predetermines the search for manifestations of cytoprotective action. Derivatives of 3-(arylmethylamino)pyridone in preliminary experiments demonstrated some potential for antioxidant/antiradical activity, therefore the present attempt to clarify the combined antiradical and cytoprotective effect is logical.

## 2. Results

### 2.1. Antiradical Activity Tests

Table 1 and Table 2 contain the results of a study of the antiradical activity of eight 3-(arylmethylamino)-6-methyl-4-phenylpyridin-2(1*H*)-one compounds prepared in this study. The final value is presented as IC50 (50% reduction in DPPH radical activity). Ascorbic acid was used as a standard for antiradical activity against DPPH.

For the studied compounds, the results were obtained in the form of fixed optical density indicators and calculated IC50 (DPPH) levels in each case. The results are presented in Table 1 and each entry represents the average of three experiments.

As can be seen from the presented data, the ability to ”extinguish” DPPH is inherent in all eight compounds under study, approximately to the same extent, with a small scatter in quantitative characteristics. The antiradical activity of the studied compounds is comparable to that of ascorbic acid.

In a similar mode, the study of the antiradical activity of the compounds with respect to ABTS•+ [2,2′-azinobis(3-ethylbenzothiazoline-6-sulfonic acid) cation radical] was carried out. As a standard of antiradical activity, vitamin E (Trolox) is present in the form of Trolox equivalent of antioxidant capacity (TEAC) (Table 2).

Evaluating the antiradical effect of the presented samples in relation to the ABTS•+ radical, it can be seen that the studied compounds have a moderate potential for antiradical activity, approaching in terms of the severity of the quantitative assessment the indicators inherent in vitamin E (Trolox).

The results presented in Table 1 and Table 2 prove that 3-(arylmethylamino)pyridone derivatives have an antiradical potential. Compounds 3a–e and 3h are not inferior to ascorbic acid in terms of the strength of the antiradical effect with respect to the DPPH radical. Compounds 3f and 3g have an antiradical potential with respect to the DPPH radical, but less than that of ascorbic acid. Compounds 3b and 3d are comparable in their antiradical effect against the ABTS radical with ascorbic acid and vitamin E (Trolox). Compounds 3a, 3e, and 3h are inferior to ascorbic acid and Trolox in terms of the strength of the antiradical effect against the ABTS radical.

### 2.2. Cytoprotective Activity Tests

In accordance with the aim of the study, the next step after establishing the antiradical effect of the investigated 3-(arylmethylamino)pyridone compounds was to study their probable cytoprotective effect. The results of measuring the level of viability of transplanted MCF-7 (Michigan Cancer Foundation-7) cells during incubation with the studied 3-(arylmethylamino)pyridine compounds are presented in Table 3.

Table 3 shows the indicators of cell viability after incubation with 3-(arylmethylamino)pyridone compounds as a percentage relative to the cell line MCF-7 taken as 100% viability in the control (cells without the addition of test compounds).

As Table 3 shows, when cells are incubated with 3e, cell viability is approximately halved. However, the incubation of MCF-7 cells with 3h and 3d gives the opposite result: these compounds apparently prevent cell damage during the 24-h incubation period and after a day their viability significantly exceeds the viability of cells in the control. It can be considered proven that samples 3h and 3d have a pronounced cytoprotective activity under the conditions of our experiment, contributing to a better survival of the MCF-7 cell line. 

For the compound with the most pronounced ability to protect cells under conditions of cultivation of the MCF-7 monolayer during the 24-h period, namely compound 3d, additional studies were performed to confirm the level of viability of MCF-7 cells in the test with neutral red. This dye is known to selectively stain cells with fully functionally preserved membranes. The results of the study of changes in the viability of MCF-7 cells under in vitro conditions under the influence of the studied samples in the test with neutral red are presented in Table 4. The table shows the indicators of cell viability as a percentage relative to the MCF-7 cell line taken as 100% viability in the control (cells without addition of test compound).

As can be seen, the results of the interaction of cells pre-incubated with the **3d** compound with the neutral red dye, which is able to selectively stain the cell only with a functionally and structurally preserved cell membrane, indicate that the cell viability after contact with the **3d** compound was significantly increased.

In experiments on the study of hemorheological activity of samples (Table 5), it was found that incubation of blood for 60 min at a temperature of 43.0 °C led to a significant increase in blood viscosity at various spindle speeds from 2 to 60 rpm, which indicates the formation of blood hyperviscosity.

### 2.3. Molecular Docking and Drug-Likeness Properties Evaluation

Currently, modern cytoprotectors are successfully used in the treatment of cerebral hypoxia of various etiologies, as well as in acute and chronic cardiovascular pathology [14,15,16]. It is known that oxidative stress is considered one of the important mechanisms of cell cytotoxicity. In medical practice, there is a sufficient arsenal of effective cytoprotectors from the group of pharmacological agents with different mechanisms of action. All of them protect cells from cytotoxic effects of various etiologies [17,18,19,20]. For example, a cell dies as a result of exposure to highly reactive oxygen radicals that destroy all types of macromolecules (RNA, DNA, proteins, lipids). At the same time, antioxidants that neutralize harmful oxygen radicals have a cytoprotective effect on cells [21,22].

In this regard, to evaluate a possible biological target and confirm the previously obtained results of the cytoprotective activity of compounds 3a–h, we used the method of molecular docking. The oxidoreductase enzyme receptor (PDB identifier: 4KEW) [23], which is directly involved in the catalytic processes of biological oxidation, was chosen as the target protein.

Three-dimensional (3D) structures were obtained from the RCSB Protein Data Bank [24], while the ligand molecules were plotted using ChemBio3D Ultra 14.0. The protein structure was prepared for docking by removing a water molecule and a native ligand and adding polar hydrogen atoms, and then converted to pdbqt format using the AutoDock MGL software package [25]. The docking process was carried out using AutoDock Vina [26]. For the oxidoreductase enzyme receptor (PDB: 4KEW) [23], active site grid coordinates (X = 14.1084, Y = 19.9038, and Z = 9.73737; size 26 × 20 × 20 Å) were used. The interaction of ligands at binding sites was interpreted using the Discovery Studio Visualizer [27].

The docking results showed (Table 6) that for the studied structures, the free energies (kcal/mol) of complexes with the selected receptors were higher than the free energy of the complex of this protein with the corresponding native ligand (comparison drug omeprazole), except for compound **3a**.

For compounds **3a**–**h**, the number of intermolecular hydrogen bonds, the binding energies of the stable ligand–receptor complex (4KWE), and the number of nearest amino acid residues were determined (Table 7).

Analysis of the interactions between the 4KWE protein complex and ligands 3a–h showed that all the studied derivatives formed fairly strong complexes with the target receptor protein (Figure 1, Figure 2, Figure 3 and Figure 4).

Thus, the results of computer docking testify to their potential cytoprotective activity. In this case, the possible presence of both acceptor and donor substituents in the methylaryl fragment increases the affinity for the selected receptor protein.

### 2.4. Compound Screening Using Lipinski’s Rule of Five

To model the bioavailability of new physiologically active substances for their possible use as an orally active drug, one of the ways to identify compounds with good absorption capacity and moderate lipophilicity is to screen compounds using Lipinski’s rule of five (RO5). The rule states that good absorption or penetration of a drug is possible if the chemical structure of the drug does not violate more than one of the following criteria/rules:(1)Molecular weight less than or equal to 500;(2)LogP (octanol-water partition coefficient) must be less than or equal to 5;(3)Hydrogen bond donors must be less than or equal to 5;(4)There should not be more than 10 hydrogen bond acceptors [28].

The Molinspiration online tool [29] was used to assess the selected compounds **3a**–**h** with potential cytoprotective activity as the likelihood of their use in medical practice.

The results obtained are shown by us in Table 8.

The results of the computer screening indicate that none of the studied compounds 3a–h violated any of the RO5 rules, indicating their possible use as drugs with antiradical and cytoprotective activity.

## 3. Discussion

The results of the MTT test and the neutral red test can be considered evidence of the presence of cytoprotective activity in 3-(arylmethylamino)pyridone compounds in the compound codenamed 3d. As was determined in the same series of experiments, the investigated compounds of 3-(arylmethylamino)pyridone compounds, including 3d, have a pronounced potential for antiradical activity against the standard DPPH and ABTS•+ tests. The data obtained allow us to suggest a potential relationship between the presence of antiradical activity and manifestations of a cytoprotective effect. The results obtained under the conditions of our experiment support the hypothesis of the possible protection of cellular and intracellular membranes of eukaryotic cells from the damaging effects of free radicals by substances that bind and neutralize free radicals [30]. In particular, one can confidently judge the relationship between antiradical and cytoprotective actions under conditions of cellular aging [31].

An absolute limitation of this study is the lack of comparison of antiradical and cytoprotective effects in vivo using adequate models. This is what is expected to be done in the future.

Thus, for 3-(arylaminomethyl)pyridone compounds, an antiradical effect on free radicals DPPH and ABTS•+ has been proven, and the results of in vitro studies on cell culture have been obtained, suggesting the possibility of a cytoprotective effect. Both results obtained under the conditions of this experiment deserve, in our opinion, detailed consideration in the forthcoming extended experiments.

## 4. Materials and Methods

IR spectra were recorded on an Infralum FT-801 spectrometer for KBr pellets. ^1^H and ^13^C NMR spectra were recorded on a Bruker DRX400 (400 and 100 MHz, respectively) and Bruker AVANCE 500 (500 and 125 MHz, respectively) instruments using DMSO-*d*_6_ (compounds **2b**, **2e**–**g**, and **3g**), DMSO-*d*_6_ + CF_3_COOH (compound **2d**), or CDCl_3_ (remaining compounds) with TMS as internal standard. Elemental analysis was performed on a Carlo Erba 1106 CHN instrument. Melting points were determined using a Koffler hot bench. Monitoring of the reaction course and the purity of the products was carried out by TLC on Sorbfilplates and visualized using iodine vapor or UV light.

### 4.1. Synthesis of Compounds

3-Amino-6-methyl-4-phenylpyridine-2(1*H*)-one **1** was synthesized according to procedure [1]. 

In order to create a library of 3-(arylaminomethyl)pyridone derivatives and then evaluate them for possible antiradical and cytoprotective activity, we performed the synthesis of compounds **3a**–**h** based on 3-amino-6-methyl-4-phenylpyridine-2(1*H*)-one **1**. 3-(Arylmethylamino)pyridones **3a**–**g** were obtained in good yields by reacting pyridine-2-one **1** with a series of aromatic aldehydes followed by reduction of imines **2a**–**g** with sodium borohydride in 2-propanol at 25–35 °C (Figure 1), their physicochemical and spectral characteristics were according to the literature procedure [10,11]. NMR Spectral Data see Appendix A.

Typical procedure for the synthesis of imines **2a**–**h**. The mixture of 3-amino-pyridin-2(1*H*)-one **1** (200 mg, 1 mmol), aromatic aldehyde (1.2 mmol), and a catalytic amount of formic acid in 5 mL of 2-propanol was refluxed for 1–3 h. After cooling the reaction mixture, precipitated imines **2a**–**h** were filtered off and washed with hexane.

Typical procedure for the synthesis of compounds **3a**–**h**. To a suspension of imines **2a**–**h** (1 mmol) in 2-propanol (15 mL) were added water (3 mL) and sodium borohydride (0.380 g, 10 mmol) with stirring at a temperature of 25–35 °C; the reaction mixture was stirred for 10–15 h. Then the reaction mixture was poured into a beaker with ice-cold water (150 mL). The aqueous layer was extracted with chloroform (3 × 25 mL), the organic layer was dried over Na_2_SO_4_, the solvent was removed by distillation, and the residue was triturated with hexane. The crude product was recrystallized from a 1:2 mixture of 2-propanol and hexane.

(*E*)-3-(Benzylideneamino)-6-methyl-4-phenylpyridin-2(1*H*)-one (**2a**). Yield: 236 mg (82%), yellow crystals, m.p.: 214–215 °C (2-propanol - chloroform). IR (KBr, cm^−1^): 3450, 3352, 1636. ^1^H NMR (400 MHz, CDCl_3_) δ ppm 2.41 (s, 3H), 6.25 (s, 1H), 7.35–7.40 (m, 5H), 7.48–7.53 (m, 3H), 7.75 (dd, ^3^*J* = 7.1 Hz, ^4^*J* = 2.4 Hz, 2H), 9.35 (s, 1H), 13.18 (br. s, 1H). ^13^C NMR (100 MHz, CDCl_3_) δ ppm 18.7, 108.7, 121.6, 127.5, 128.0, 128.4, 128.6, 130.0, 130.6, 131.0, 131.7, 137.7, 140.8, 146.3, 161.2, 162.6. Anal. Calcd. for C_19_H_16_N_2_O: C, 79.14; H, 5.59; N, 9.72. Found: C, 79.52; H, 5.87; N, 9.91.

(*E*)-3-((2-Hydroxybenzylidene)amino)-6-methyl-4-phenylpyridin-2(1*H*)-one (**2b**). Yield: 289 mg (95%), yellow crystals, m.p.: 251–252 °C (2-propanol). IR (KBr, cm^−1^): 3454, 2826, 1626, 1466, 1282. ^1^H NMR (400 MHz, DMSO-*d*_6_) δ ppm 2.24 (s, 3H, CH_3_); 6.13 (s, 1H, H-5); 6.71 (d, 1H, ^3^*J* = 8.2 Hz, H-3’ Ar); 6.86 (td, 1H, ^3^*J* = 7.4 Hz, ^4^*J* = 1.0 Hz, H-5’ Ar); 7.26 (td, 1H, ^3^*J* = 7.7 Hz, ^4^*J* = 1.5 Hz, H-4’ Ar); 7.36–7.44 (m, 6H, 5H Ph, H-6’ Ar); 9.86 (s, 1H, =C-H); 12.16 (s, 1H, NH); 12.59 (s, 1H, OH). ^13^C NMR (100 MHz, DMSO-*d*_6_) δ ppm 18.3, 107.1, 116.3, 118.8, 119.7, 127.9, 128.27, 128.31, 128.5, 132.3, 132.4, 137.9, 142.8, 147.2, 159.1, 159.9, 164.3. Anal. Calcd. for C_19_H_16_N_2_O_2_: C, 75.31; H, 5.66; N, 9.57. Found: C, 74.98; H, 5.30; N, 9.20.

(*E*)-3-((4-Methoxybenzylidene)amino)-6-methyl-4-phenylpyridin-2(1*H*)-one (**2c**). Yield: 261 mg (82%), yellow crystals, m.p.: 217–219 °C (2-propanol- chloroform). IR (KBr, cm^−1^): 2901, 1625, 1574, 1248. ^1^H NMR (400 MHz, CDCl_3_) δ ppm 2.40 (s, 3H, CH_3_); 3.83 (s, 3H, OCH_3_); 6.23 (d, 1H, ^4^*J* = 0.8 Hz, H-5); 6.90 (d, 2H, ^3^*J* = 8.8 Hz, H-3,5 Ar); 7.38 (d, 2H, ^3^*J* = 7.4 Hz, H-2,6 Ph); 7.51 (td, 3H, ^3^*J* = 8.7 Hz, ^4^*J* = 1.5 Hz, H-3,4,5 Ph); 7.70 (d, 2H, ^3^ J =8.8 Hz, H-2,6 Ar); 9.40 (s, 1H, N=CH); 13.05 (br. s, 1H, NH). ^13^C NMR (100 MHz, CDCl_3_) δ ppm 18.7 (CH_3_), 55.3 (OCH_3_), 108.1, 113.9 (2C, Ar), 127.5 (2C, Ph), 128.1 (2C, Ph), 128.9, 130.0 (2C, Ar), 132.2, 138.1, 140.2, 145.4, 159.9, 161.3, 161.8, 162.0. Anal. Calcd. for C_20_H_18_N_2_O_2_: C, 75.45; H, 5.70; N, 8.80. Found: C, 75.04; H, 6.06; N, 8.38.

(*E*)-6-Methyl-3-((3-nitrobenzylidene)amino)-4-phenylpyridin-2(1*H*)-one (**2d**). Yield: 260 mg (78%), orange crystals, m.p.: 290–291 °C. IR (KBr, cm^−1^): 2893, 1655, 1624, 1534. ^1^H NMR (400 MHz, DMSO-*d*_6_ + CF_3_COOH) δ ppm 2.16 (s, 3H, CH_3_); 5.98 (s, 1H, H-5); 7.38 (s, 5H, Ph); 7.70 (m, 1H, H-6 Ar); 8.17 (m, 1H, H-5 Ar); 8.36 (m, 1H, H-6 Ar); 8.55 (s, 1H, H-2 Ar); 9.99 (br. s, 1H, N=CH); 13.18 (br. s, 1H, NH). ^13^C NMR (100 MHz, DMSO-*d*_6_) δ ppm 19.5 (CH_3_), 108.2 (C-5), 116.9, 125.1, 129.0, 129.5 (2C Ph), 129.7, 130.4 (2C Ph), 130.9, 131.9, 136.1, 136.3, 138.8, 146.4, 148.2, 149.9, 192.3. Anal. Calcd. for C_19_H_15_N_3_O_3_: C, 68.46; H, 4.54; N, 12.61. Found: C, 68.88; H, 4.93; N, 12.18.

(*E*)-3-((3,4-Dimethoxybenzylidene)amino)-6-methyl-4-phenylpyridin-2(1*H*)-one (**2e**). Yield: 294 mg (84%), yellow crystals, m.p.: 220–224 °C. IR (KBr cm^−1^): 2831, 1605, 1583, 1271. ^1^H NMR (500 MHz, DMSO-*d*_6_) δ ppm 2.21 (s, 3H, CH_3_); 3.79 (s, 3H, OCH_3_); 3.84 (s, 3H, OCH_3_); 6.13 (s, 1H, H-5); 6.53 (dd, 1H, ^3^*J* = 8.7 Hz, ^5^*J* = 2.1 Hz, H-5 Ar); 6.60 (d, 1H, ^5^*J* = 2.2 Hz, H-2 Ar); 7.33–7.38 (m, 3H, H-3,4,5 Ph); 7.42–7.43 (m, 2H, H-2,6 Ph); 7.64 (d, 1H, ^3^*J* = 8.8 Hz, H-6 Ar); 9.54 (s, 1H, N=CH); 11.81 (br. s, 1H, NH). ^13^C NMR (125 MHz, DMSO-*d*_6_) δ ppm 18.3 (CH_3_), 55.4 (OCH_3_), 55.6 (OCH_3_), 98.0 (C-2, Ar), 106.2 (C-5), 106.5 (C-5 Ar), 118.4 (C-6 Ar), 127.4 (2C, Ph), 127.5, 127.7, 129.7 (2C, Ph), 132.1, 138.2, 140.2, 143.2, 155.5, 159.2, 160.4, 163.1. Anal. Calcd. for C_21_H_20_N_2_O_3_: C, 72.40; H, 5.79; N, 8.04. Found: C, 72.05; H, 6.21; N, 8.39.

(*E*)-3-((2,4-Dimethoxybenzylidene)amino)-6-methyl-4-phenylpyridin-2(1*H*)-one (**2f**). Yield: 296 mg (85%), yellow crystals, m.p.: 241–244 °C. IR (KBr cm^−1^): 2838, 1635, 1621, 1262, 1230. ^1^H NMR (500 MHz, DMSO-*d*_6_) δ ppm 2.23 (s, 3H, CH_3_); 3.68 (s, 3H, OCH_3_); 3.78 (s, 3H, OCH_3_); 6.19 (s, 1H, H-5); 6.79 (d, 1H, ^3^*J* = 8.3 Hz, H-5 Ar); 7.22 (dd, 1H, ^3^*J* = 8.2 Hz, ^5^*J* = 1.6 Hz, H-6 Ar); 7.28 (d, 1H, ^5^*J* = 1.5 Hz, H-3 Ar); 7.34 (t, 1H, ^3^*J* = 7.2 Hz, H-4 Ph); 7.39 (t, 2H, ^3^*J* = 7.3 Hz, H-3,5 Ph); 7.48 (d, 2H, ^3^*J* = 7.1 Hz, H-2,6 Ph); 9.41 (s, 1H, N=CH); 11.91 (br. s, 1H, NH). ^13^C NMR (125 MHz, DMSO-*d*_6_) δ ppm 18.3 (CH_3_), 55.1 (OCH_3_), 55.6 (OCH_3_), 106.6 (C-3 Ar), 109.0 (C-5), 111.3 (C-5 Ar) 123.0 (C-6 Ar), 127.4 (2C Ph), 127.8, 129.9 (2C Ph), 130.6, 130.7, 138.1 140.1, 144.4, 148.9, 151.2, 159.3, 160.0. Anal. Calcd. for C_21_H_20_N_2_O_3_: C, 72.40; H, 5.79; N, 8.04. Found: C, 72.08; H, 6.18; N, 8.41.

(*E*)-3-((5-Bromo-2-hydroxybenzylidene)amino)-6-methyl-4-phenylpyridin-2(1*H*)-one (**2g**). Yield: 362 mg (94%), yellow crystals, m.p.: 294–297 °C. IR (KBr, cm^−1^): 2922, 1625, 1621, 1162, 820. ^1^H NMR (400 MHz, DMSO-*d*_6_) δ ppm 2.27 (s, 3H, CH_3_); 6.14 (s, 1H, H-5); 6.69 (d, 1H, ^3^*J* = 9.2 Hz, H-3 Ar); 7.36–7.44 (m, 6H, H-4 Ar, H-2,3,4,5,6 Ph); 7.63(s, 1H, H-6 Ar) 9.83 (s, 1H, N=CH); 12.03 (br. s, 1H, NH); 12.50 (s, 1H, OH). Anal. Calcd. for C_19_H_15_BrN_2_O_2_: C, 59.55; H, 3.95; N, 7.31. Found: C, 59.98; H, 4.38; N, 7.70.

(*E*)-3-((4-Dimethylamino)benzylidene)amino)-6-methyl-4-phenylpyridin-2(1*H*)-one (**2h**). The mixture of 3-aminopyridin-2(1*H*)-one **1** (200 mg, 1 mmol), 654 mg 4-(dimethylamino)benzaldehyde (43 mmol) and catalytic amount of formic acid in 30 mL 2-propanol was refluxed for 8 h. After cooling the reaction mixture, precipitated imine **2h** was filtered off and washed with hexane. Yield: 310 mg (93%), yellow crystals, m.p.: 258–261 °C (2-propanol). IR (KBr): ν, cm^−1^: 3748, 2894, 1633, 1614. ^1^H NMR (400 MHz, CDCl_3_) *δ* ppm 2.36 (s, 3H, CH_3_); 3.01 (s, 6H, N(CH_3_)_2_); 6.20 (s, 1H, H-5); 6.66 (d, 2H, *J* = 7.3, H-3’,5’ Ar); 7.33–7.36 (m, 3H, H-3,4,5 Ph); 7.51 (d, 2H, *J* = 7.3, H-2,6 Ph); 7.63 (d, 2H, *J* = 7.3, H-3’,5’ Ar); 9.20 (s, 1H, N=CH); 12.55 (br. s, 1H, NH). ^13^C NMR (100 MHz, CDCl_3_) *δ* ppm 18.8 (CH_3_); 40.2 (N(CH_3_)_2_) 108.5 (C-5); 111.4 (C-3’,5’ Ar); 125.7; 127.5 (C-3,5 Ph); 127.7 (C-4 Ph); 130.1 (C-2’,6’ Ar); 130.2 (C-2,6 Ph); 133.3 (C-1’ Ar); 138.2; 139.1; 143.9; 152.2; 161.3 (C-2); 162.9 (N=CH). Anal. Calcd. for C_21_H_21_N_3_O: C, 76.11; H, 6.39; N, 12.68. Found: C, 75.74; H, 6.81; N, 12.28.

3-(Benzylamino)-6-methyl-4-phenylpyridin-2(1*H*)-one (**3a**). Yield: 235 mg (81%), yellow crystals, m.p.: 167–169 °C (2-propanol-hexane). IR (KBr, cm^−1^): 3334, 3274, 2925, 1831, 1628. ^1^H NMR (400 MHz, CDCl_3_) δ ppm 2.27 (s, 3H, CH_3_); 3.73 (s, 2H, N-CH_2_); 5.19 (br. s, 1H, N-H); 5.87 (s, 1H, H-5); 6.88 (m, 2H, H-2’,6’ Ar); 7.0 (m, 3H, H-3’,4’,5’ Ar); 7.3 (m, 5H, Ph); 12.38 (s, 1H, NHCO); ^13^C NMR (100 MHz, CDCl_3_) δ ppm 18.2; 49.9; 110.6; 126.5; 127.3; 127.6; 128.0; 128.2; 128.3; 132.2; 132.6; 132.9; 139.0; 139.9; 160.2. Anal. Calcd. for C_19_H_18_N_2_O: C, 78.59; H, 6.25; N, 9.65; Found: C, 74.91; H, 6.05; N, 9.86.

3-((2-Hydroxybenzyl)amino)-6-methyl-4-phenylpyridin-2(1*H*)-one (**3b**). Yield: 217 mg (71%), yellow crystals, m.p.: 187–188 °C (2-propanol-hexane). IR (KBr, cm^−1^): 3367, 2920, 1632, 1542. ^1^H NMR (400 MHz, CDCl_3_) δ ppm 2.47 (s, 3H, CH_3_); 3.62 (bt, 1H, *J* = 7.6 Hz, N-H); 4.24 (d, 2H, *J* = 6.9 Hz, N-CH_2_); 6.10 (s, 1H, H-5); 6.74 (td, 1H, *J* = 7.3 Hz, *J* = 1.4 Hz, H-5’ Ar); 6.91 (dd, 1H, ^3^*J* = 8.2 Hz, ^4^*J* = 0.9 Hz H-3’ Ar), 6.99 (dd, 1H, *J* = 7.3 Hz, *J* = 1.4 Hz, H-6’ Ar), 7.18 (td, 1H, *J* = 7.8 Hz, *J* = 1.4 Hz, H-4’ Ar), 7.37–7.43 (m, 3H, H-3,4,5 Ph), 7.47 (d, 2H, *J* = 7.1 Hz, H-2,6 Ph), 10.05 (s, 1H, OH); 13.38 (s, 1H, NHCO). ^13^C NMR (100 MHz, CDCl_3_) δ ppm 18.6, 48.8, 109.5, 116.7, 118.5, 124.4, 127.8, 128.5, 129.2, 129.3, 131.8, 137.2, 137.4, 140.9 157.5, 162.7. Anal. Calcd. for C_19_H_18_N_2_O_2_: C, 74.49; H, 5.92; N, 9.14; Found: C, 74.81; H, 5.74; N, 9.26.

3-((4-Methoxybenzyl)amino)-6-methyl-4-phenylpyridin-2(1*H*)-one (**3c**). Yield: 176 mg (55%), brown crystals, m.p.: 154–157 °C (2-propanol-hexane). IR (KBr, cm^−1^): 3303, 2838, 1884, 1630, 1509. ^1^H NMR (400 MHz, CDCl_3_) δ ppm 2.27 (s, 3H, CH_3_); 3,74 (s, 3H, OCH_3_) 3.75 (s, 2H, N-CH_2_); 5.00 (br. s, 1H, N-H); 5.91 (s, 1H, H-5); 6.74 (d, 2H, ^3^*J* = 9.2 Hz, H-3,5 Ar); 6.97 (d, 2H, ^3^*J* = 6.1 Hz, H-2,6 Ar); 7.32 (t, 1H, ^3^*J* = 6.9 Hz, H-4 Ph); 7.38–7.41 (m, 2H, H-5,3 Ph); 7.45–7.47 (m, 2H, H-2,6 Ph); 12.77 (s, 1H, NH). ^13^C NMR (100 MHz, CDCl_3_) δ ppm 18.3 (CH_3_), 49.8 (N-CH_2_), 55.2 (OCH_3_), 109.4 (C-5), 113.5 (C-3,5 Ar), 127.5, 128.3 (4C Ph), 128.7 (C-2,6 Ar), 132.1, 132.3, 133.0, 139.3, 158.4, 161.6. Anal. Calcd. for C_20_H_20_N_2_O_2_: C, 74.98; H, 6.29; N, 8.74; Found: C, 74.54; H, 6.66; N, 8.34.

6-Methyl-3-((3-nitrobenzyl)amino)-4-phenylpyridin-2(1*H*)-one (**3d**). Yield: 255 mg (76%), orange crystals, m.p.: 160–163 °C (2-propanol). IR (KBr, cm^−1^): 3359, 3303, 2923, 1844, 1642. ^1^H NMR (400 MHz, CDCl_3_) δ ppm 2.28 (s, 3H, CH_3_); 3.96 (d, 2H, ^3^*J* = 6.1 Hz, N-CH_2_); 5.19 (br. s, 1H, N-H); 5.89 (s, 1H, H-5); 7.32–7.36 (m, 3H, H-3,4,5 Ph); 7.37–7.38 (m, 4H, H-2,6 Ph, H-5,6 Ar) 7.80 (s, 1H, H-2 Ar); 8.01 (d, 1H, ^3^*J* = 6.1 Hz, H-4Ar) 12.74 (s, 1H, NH). ^13^C NMR (100 MHz, CDCl_3_) δ ppm 18.3 (CH_3_), 49.2 (N-CH_2_), 109.4 (C-5), 121.9 (C-4 Ar), 122.4 (C-2 Ar), 127.9, 128.2 (2C Ph), 128.3 (2C Ph), 129.0, 131.9, 132.3, 132.9, 133.4, 142.4, 148.0, 161.5. Anal. Calcd. for C_19_H_17_N_3_O_3_: C, 68.05; H, 5.11; N, 12.53; Found: C, 68.48; H, 5.52; N, 12.93.

3-((3,4-Dimethoxybenzyl)amino)-6-methyl-4-phenylpyridin-2(1*H*)-one (**3e**). Yield: 179 mg (51%), yellow crystals, m.p.: 132–133 °C (2-propanol-hexane). IR (KBr, cm^−1^): 3341, 2931, 1629, 1518. ^1^H NMR (400 MHz, CDCl_3_) δ ppm 2.26 (s, 3H, CH_3_); 3.75 (s, 2H, N-CH_2_); 3,78 (s, 3H, OCH_3_); 3,82 (s, 3H, OCH_3_); 5.04 (br. s, 1H, N-H); 5.90 (s, 1H, H-5); 6.56 (s, 1H, H-2 Ar); 6.57 (d, 1H, ^3^*J* = 7.4 Hz, H-5 Ar); 6.70 (d, 1H, ^3^*J* = 7.3 Hz, H-6 Ar); 7.32 (t, 1H, ^3^*J* = 7.3 Hz, H-4 Ph); 7.39 (t, 2H, ^3^*J* = 7.9 Hz, H-3,5 Ph); 7.43 (m, 2H, H-2,6 Ph); 12.74 (br. s, 1H, NH). ^13^C NMR (100 MHz, CDCl_3_) δ ppm 18.3 (CH_3_), 50.0 (N-CH_2_), 55.6 (OCH_3_), 55.8 (OCH_3_), 109.4 (C-5), 110.7 (C-2 Ar), 110.7 (C-5 Ar), 119.5 (C-6 Ar), 127.5, 128.3 (2C Ph), 128.3 (2C Ph), 131.6, 132.1, 132.8, 133.0, 139.2, 147.7, 148.6, 161.6. Anal. Calcd. for C_21_H_22_N_2_O_3_: C, 71.98; H, 6.33; N, 7.99; Found: C, 72.39; H, 6.72; N, 8.36.

3-((2,4-Dimethoxybenzyl)amino)-6-methyl-4-phenylpyridin-2(1*H*)-one (**3f**). 

Yield: 214 mg (61%), yellow crystals, m.p.: 181–184 °C (2-propanol-hexane). IR (KBr, cm^−1^): 3304, 2938, 1865, 1644. ^1^H NMR (400 MHz, CDCl_3_) δ ppm 2.25 (s, 3H, CH_3_); 3,64 (s, 3H, OCH_3_); 3,75 (s, 3H, OCH_3_) 3.84 (s, 2H, N-CH_2_); 5.07 (br. s, 1H, N-H); 5.88 (s, 1H, H-5); 6.30 (d, 1H, ^3^*J* = 7.7 Hz, H-5 Ar); 6.32 (s, 1H, H-3 Ar); 6.81 (d, 1H, ^3^*J* = 7.6 Hz, H-6 Ar); 7.31 (t, 1H, ^3^*J* = 6.9 Hz, H-4 Ph); 7.36–7.43 (m, 4H, H-2,3,5,6 Ph); 12.54 (s, 1H, NH). ^13^C NMR (100 MHz, CDCl_3_) δ ppm 18.3 (CH_3_), 45.0 (N-CH_2_), 55.0 (4-OCH_3_), 55.2 (2-OCH_3_), 98.2 (C-3 Ar), 103.3 (C-5), 109.2 (C-5 Ar), 120.8, 127.3, 128.3 (4C Ph), 129.8, 131.7, 131.9, 133.3, 139.4, 154.5, 159.9, 161.6. Anal. Calcd. for C_21_H_22_N_2_O_3_: C, 71.98; H, 6.33; N, 7.99; Found: C, 71.54; H, 6.75; N, 8.42.

3-((5-Bromo-2-hydroxybenzyl)amino)-6-methyl-4-phenylpyridin-2(1*H*)-one (**3g**). Yield: 285 mg (74%), yellow crystals, m.p.: 203–206 °C (2-propanol-hexane). IR (KBr, cm^−1^): 3370, 3279, 1842, 1643. ^1^H NMR (400 MHz, DMSO-*d*_6_) δ ppm 2.09 (s, 3H, CH_3_); 3.66 (d, *J* = 7.6 Hz, 2H, N-CH_2_); 5.10 (bt, 1H, ^3^*J* = 6.9 Hz, N-H); 5.81 (s, 1H, H-5); 6.62 (d, 1H, ^3^*J* = 9.2 Hz, H-3 Ar); 6.83 (br. s, 1H, H-6 Ar); 7.11–7.14 (dd, 1H, ^3^*J* = 9.2 Hz, ^4^*J* = 3.1 Hz, H-4 Ar); 7.34 (td, 1H, ^3^*J* = 6.1 Hz, ^4^*J* = 3.1 Hz, H-4 Ph); 7.39–7.42 (m, 4H, H-3,2,5,6 Ph); 9.8 (s, 1H, OH); 11.6 (s, 1H, NH). ^13^C NMR (100 MHz, DMSO-*d*_6_) δ ppm 17.8 (CH_3_), 44.5 (N-CH_2_), 107.5 (C-5 Ar), 109.5 (C-5), 116.9 (C-4 Ar), 127.5 (C-3 Ar), 127.9 (2C Ph), 128.5 (2C Ph), 128.9, 129.1, 130.3 (C-6 Ar), 130.9, 132.0, 132.6, 139.0, 154.7, 160.1. Anal. Calcd. for C_19_H_17_BrN_2_O_2_: C, 59.23; H, 4.45; N, 7.27; Found: C, 59.63; H, 4.84; N, 7.70.

3-((4-Dimethylaminobenzyl)amino)-6-methyl-4-phenylpyridin-2(1*H*)-one (**3h**). 

To a suspension of imine **2h** (123 mg, 3.7 mmol) in 2-propanol (50 mL) was added water (3 mL) and sodium borohydride (112 mg, 30 mmol) with stirring at a temperature of 25–35 °C; the reaction mixture was stirred for 10 h. Then the reaction mixture was poured into a beaker with ice-cold water (150 mL). The aqueous layer was extracted with chloroform (3×25 mL), the organic layer was dried over Na_2_SO_4_, the solvent was removed by distillation, and the residue was triturated with hexane. The crude product was recrystallized from a 1:2 mixture of 2-propanol and hexane. Yield: 250 mg (75%), yellow crystals, m.p.: 165–168 °C (2-propanol-hexane). IR (KBr, cm^−1^): 2922, 1643, 1518. ^1^H NMR (400 MHz, CDCl_3_) *δ* ppm 2.27 (s, 3H, CH_3_); 2.89 (s, 6H, N(CH_3_)_2_); 3.70 (s, 2H, N-CH_2_); 4.97 (br. s, 1H, N-H); 5.93 (s, 1H, H-5); 6.60 (d, 2H, *J* = 9.2, H-3’,5’ Ar); 6.95 (d, 2H, *J* = 8.5, H-3’,5’ Ar); 7.32 (t, 1H, *J* = 7.3, H-4 Ph); 7.41 (t, *J* = 7.3, 2H, H-3,5 Ph); 7.49 (d, 2H, *J* = 7.3, H-2,6 Ph); 12.83 (br. s, 1H, NH). ^13^C NMR (100 MHz, CDCl_3_) *δ* ppm 18.3 (CH_3_); 40.7 (N(CH_3_)_2_); 50.0 (CH_2_-Ar); 109.4 (C-5); 112.5 (C-3’,5’ Ar); 127.4 (C-4 Ph); 128.2; 128.2 (C-2’,6’ Ar); 128.3 (C-2,6 Ph); 128.5 (C-3,5 Ph); 131.0; 131.9; 133.4; 139.5; 149.6 (C-4’ Ar); 161.6 (C-2). Anal. Calcd. for C_21_H_23_N_3_O: C, 75.65; H, 6.95; N, 12.60; Found: C, 75.22; H, 5.65; N, 9.94.

### 4.2. DPPH and ABTS Radical Scavenging Assay

The antiradical action of the compounds **3h**, **3e**, and **3d** was routinely studied in relation to the radicals of 2,2-diphenyl-1-picrylhydrazyl (DPPH) and 2,2’-azinobis(3-ethylbenzothiazoline-6-sulfonic acid) (ABTS•+) [32].

To assess the antiradical activity of the studied samples (**3h**, **3e**, and **3d**), the standard DPPH test purchased from Sigma Aldrich, Steinheim, Germany was used in accordance with the manufacturer’s instructions. To select substances with pronounced antiradical activity, 2 mL of a 100 μM ethanol solution of DPPH were mixed with 20 μL of the test samples dissolved in methanol at a concentration of 5 mM. As a result, the final concentration of the test substance in the reaction mixture was 50 µM. Ten minutes after adding the test compound solution to the DPPH radical solution, the decrease in absorbance at 515 nm was measured. For substances capable of reducing optical density by more than 50%, a test was performed for interaction with the DPPH radical at final concentrations of the studied substances of 50, 25, 20, 15, 10, 5, and 2.5 µM. After that, the concentration of the test substance capable of reducing the optical density by 50% was determined—IC50 (DPPH). As a reference drug, a substance with known antiradical properties, ascorbic acid, was used.

The antiradical activity of the presented samples was studied against the radical cation 2,2’-azinobis(3-ethylbenzothiazoline-6-sulfonic acid) (ABTS•+) using the Antioxidant Assay Kit (Sigma Aldrich). The principle of the method is the formation of a ferryl myoglobin radical from metmyoglobin and hydrogen peroxide, which oxidizes ABTS with the formation of a cation radical: ABTS•+. ABTS•+ is a stable radical that can exist in aqueous solutions for quite a long time, however, the introduction of various antiradical agents into the solution leads to their interaction with ABTS•+ and rapid consumption (“quenching”) of the latter. The consumption of ABTS•+ is accompanied by characteristic spectral changes, which make it possible to record the reaction rate. The ability to dose the initial concentration of radicals in the system and control the rate of their ”quenching”" led to the widespread use of ABTS•+ to standardize the antiradical activity of various compounds [33]. We compared the rate of "quenching" of ABTS•+ by the test substances and the standard, which was used as a semi-synthetic water-soluble analogue of vitamin E, which has the commercial name Trolox. The use of Trolox allows you to evaluate the effectiveness of anti-radical action through the measure of so-called "Trolox equivalent of antioxidant efficiency": TEAC (Trolox Equivalent Antioxidant Capacity). The TEAC values indicate how much Trolox in mmol/l (mM) ”extinguishes” ABTS•+ with the same efficiency as 1 mM of the analyte. The analyzed samples (**3h** and **3e**) were dissolved in DMSO to a concentration of 10 mM, then twice 10-fold dissolved in 1×Assay Buffer to a final concentration of 0.1 mm. All test substances were tested at a concentration of 0.1 mM.

### 4.3. Cell Viability Assays

The cytoprotective properties of the presented samples (**3h**, **3e**, and **3d**) were evaluated in the MTT test on the breast cancer cell line MCF-7. Of the test sample, 1 mg was dissolved in 1 mL of DMSO. 10 μL of the dissolved substance was added to 100 μL of the nutrient medium with MCF-7 cells. Cells in nutrient medium without the addition of test compounds served as controls. After cells were incubated for 24 h with the test objects, viability was determined in the MTT test using the In Vitro Toxicology Assay Kit, MTT based (Sigma Aldrich) according to the manufacturer’s instructions. 

The In Vitro Toxicology Assay Kit Neutral Red Based (Sigma Aldrich) was used on the MCF-7 cell line according to the manufacturer’s manual to access compounds with pronounced cytoprotective activity.

### 4.4. Study of Hemorheological Effects on the Model of Blood Hyperviscosity In Vitro

Hyperviscosity syndrome (HBIS) was reproduced in vitro by blood incubation at 43.0 °C for 60 min. Blood viscosity was measured on a Brookfield DV2T rotational viscometer at various spindle speeds (from 2 to 60 rpm).

Studies of the hemorheological activity of 6 samples were carried out on 12 Wistar female rats, 12 weeks old, weighing 220–240 g. After blood sampling, the initial blood viscosity was determined in laboratory animals, and then blood samples were incubated with the test substances at a temperature of 43.0 °C for 60 min and then measured the parameters under study. The blood was incubated with the test objects dissolved in DMSO; the final concentration of the compounds was 10^−5^ g/mL of blood. Blood samples to which DMSO solvent was added in an equivolume amount served as controls. As a reference drug, a substance with known hemorheological properties, pentoxifylline, was used at a concentration of 10^−5^ g/mL of blood [34,35]. Blood incubation for 1 h under these conditions was accompanied by the formation of blood hyperviscosity [36]. The initial blood viscosity from each animal was measured once; the blood viscosity after incubation was measured in two samples from each animal, both in control and experimental samples.

### 4.5. Ethical Considerations

All research work with laboratory animals was performed in accordance with generally accepted ethical standards for the treatment of animals, based on standard operating procedures that comply with the rules adopted by the European Convention for the Protection of Vertebrate Animals used for Research and other Scientific Purposes (Strasbourg, France, 1986). The study protocol of the project “Search for means of pharmacological correction of increased blood viscosity syndrome associated with endocrine pathology” was approved on 7 August 2020 (Protocol No. 3) by the Local Ethics Commission of the Republican State Enterprise “National Center for Biotechnology” (IRB00013497 National Center of Biotechnology IRB #1).

### 4.6. Data Evaluation

Statistical processing of the results was carried out using the Excel program. The results obtained are presented as mean ± standard error of the mean.

## 5. Conclusions

Ss a result of the bioscreening performed on two tests by measuring cell viability (neutral red and MTT tests), as well as in the test under conditions of blood hyperviscosity of synthesized derivatives of 3-(arylmethylamino)-6-methyl-4-phenylpyridin-2(1*H*)-ones, it was found that they have a fairly high cytoprotective potential. It was shown that some derivatives of 3-(arylmethylamino)-6-methyl-4-phenylpyridin-2(1*H*)-ones under the conditions of our experiment have a pronounced cytoprotective activity, providing better cell survival in vitro, including in hyperviscosity conditions.

Correlation of the results obtained in vitro was performed using molecular docking on the example of the selected target protein of the oxidoreductase enzyme receptor (PDB identifier: 4KEW); the oxidoreductase enzyme receptor showed that all synthesized derivatives of 3-(arylmethylamino)-6-methyl-4-phenylpyridin-2(1*H*)-ones have a higher affinity for the selected protein than the standard gastro-cytoprotector omeprazole.

The investigated derivatives of 3-(arylmethylamino)-6-methyl-4-phenylpyridin-2(1*H*)-ones also fully satisfy Lipinski’s rule of five (RO5), which increases their chances for possible use as orally active drugs with good absorption capacity and moderate lipophilicity.

## Data Availability

Not applicable.

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
