# Peer review of "Cytoprotective Activity of Newly Synthesized 3-(Arylmethylamino)-6-Methyl-4-Phenylpyridin-2(1H)-Ones Derivatives"

_molecules, 2022, doi:10.3390/molecules27175362_

Round 1
Reviewer 1 Report
Please see the attached file below.

Author Response
Thank you for you review, we made all necessary corrections

Reviewer 2 Report
This manuscript describes cytoprotective activity of pyridin-2(1H)-one compounds. The compounds were synthesized by authors, and the biological activity of them was evaluated. In addition, docking studies are discussed. I think this kind of the series studies could attract the interest for readers. Therefore, this manuscript could be acceptable as an Article in Molecules after considering some comments as below.
“H” of 3-(arylmethyl)-6-methyl-4-phenylpryidine-2(1H)-ones should be written in italic style through the whole manuscript.
I could not find Scheme 1.
Compound numbers should be written in bold style through the whole manuscript.
Why did the authors select a–h as an aryl group at 3-position? I hope to describe the reason in the main text, if possible.
Table 6 is so complicated that I hope to describe them more simply.
In page 7 at the lune 6, why did the authors set the temperature at 43.0 °C? I hope to describe the reason in the main text, if possible.
In Table 9, how did the authors calculated miLog P? I suggest adding it anywhere in the main text.
I could not find Conclusion.
Author Response
Reviewer’s comments Answers
“H” of 3-(arylmethyl)-6-methyl-4-phenylpryidine-2(1H)-ones should be written in italic style through the whole manuscript.
Corrected throughout the manuscript
I could not find Scheme 1. Corrected
Compound numbers should be written in bold style through the whole manuscript. Corrected throughout the manuscript
Why did the authors select a–h as an aryl group at 3-position? I hope to describe the reason in the main text, if possible. added as per reviewer's note
«The primary amino group is one of the privileged reactive sites for potential modification of the obtained 4-aryl(hetaryl)-substituted 3-aminopyridine-2(1H)-ones»
Table 6 is so complicated that I hope to describe them more simply. Table 6 presents data reflecting the hemorheological effects of the studied compounds when reproducing blood hyperviscosity in vitro by incubating it at 43 ºC for 1 hour (reference to the method of reproducing blood hyperviscosity in vitro: Plotnikov, M.B.; Koltunov, A.A.; Aliyev, O.I. Method of selection of medicinal substances affecting the rheological properties of blood in vitro // Experimental and clinical pharmacology. - 1996. - No. 6. - pp. 57-58.). According to the method, the viscosity of whole blood is measured before incubation and addition of the test substance (experiment) or solvent (control) and 60 minutes after.
For each compound, baseline blood viscosities before incubation, blood viscosities 1 hour after incubation at 43 ºC without addition of test compounds, and blood viscosities 1 hour after incubation with test compounds at 43 ºC are given. Data are presented as mean ± standard error of the mean. Table 6 also lists the p values that are needed to prove that the test compounds are able to prevent the increase in blood viscosity under the conditions of this test system. All of these data are important for understanding the potential hemorheological effects of the studied compounds.
In page 7 at the lune 6, why did the authors set the temperature at 43.0 °C? I hope to describe the reason in the main text, if possible. We used the method of reproducing the model of hyperviscosity syndrome in vitro [Plotnikov, M.B.; Koltunov, A.A.; Aliyev, O.I. Method of selection of medicinal substances affecting the rheological properties of blood in vitro // Experimental and clinical pharmacology. - 1996. - No. 6.-pp. 57-58]. According to the literature data, incubation of blood for an hour at an elevated temperature (42-45°C) leads to a distinct and stable change in its rheological properties. In our experiments, we chose the optimal temperature for reproducing blood hyperviscosity in vitro, which was 43 °C.
In Table 9, how did the authors calculated miLog P? I suggest adding it anywhere in the main text. Indicated in the text, line 231
«The Molinspiration online tool [29] was used to assess the selected compounds 3a-h with potential cytoprotective activity as the likelihood of their use in medical practice»
I could not find Conclusion. added
Reviewer 3 Report
The summary must be reformulated; it is synthetic but too vague/general. The obtained results must be concretized.
Page 2, lines 60-62: the sentence must be revised, reformulated, because it has no predicate.
Table 2 must contain only the code of the tested compounds, because the molecular formula and molecular mass are mentioned in the experimental part, chapter Chemistry.
The acronym ABTSŸ+ must be explained.
The sentence on page 5, lines 117-118 must be reformulated, because the TEAC values ​​for some of the compounds (3a, 3e, 3h) are not comparable with the value obtained for ascorbic acid.
The acronyms MCF-7 and MTT must be explained.
Table 5: the expression "vitamin stained with neutral red" must be explained.
Can't the very high capacity of compound 3d to increase cell viability also have unfavorable consequences? Does it not suggest other unfavorable properties related to cell multiplication?
How do the authors interpret the compounds' ability to increase blood viscosity? Isn't it an unfavorable property, which is in detriment for the investigated compounds?
The sentence on page 9, lines 186-189 must be completed with the expression "except for compound 3a".
In figures 2-5, the 2D formulas must be redone in order to be able to distinguish between the different types of interactions signaled with shades of the same color. In addition, I do not think that a group of atoms can interact, at the same time, with several groups, by hydrogen bonds or by other types of interactions.
I have reservations about some types of interactions mentioned in the figures. Bibliographic references must be given in this sense.
The nitro group in fig. 3 is written incorrectly. Attention, the nitrogen atom has 10 electrons in the valence layer, which is impossible.
The phrase on page 13, lines 231-232 must be reformulated, because it does not seem to me that there is such a good correlation between the antiradical potential and the cytoprotective activity.]
The bands in the IR spectra must be mentioned in descending order of wave number, for example for compound 3h: 1633, 1614.
Page 14, lines 277-278: explanation for the notations "1H NMR (400 МГц, CDCl3, δ м.д.) 278"!!!!
I believe that the experimental part of chemistry should be completed with the characterization data of all compounds or at least of those used in biological experiments. Otherwise, the title must be changed, because it is misleading. It is mentioned in the title, newly synthesized compounds. Therefore, even if they are mentioned in the literature, if they were synthesized by the authors, I consider that they must be isolated, purified and characterized, to be sure that the biological tests are representative for these compounds.
Page 15, line 304: what does DPPH or DPPH mean?
Author Response
Thank you, revised manuscript is attached

Round 2
Reviewer 1 Report
Please see the attached file.

Author Response
Thank you for the review, we are uploading corrected version of manuscript, you may find the corrections in the attached manuscript

Reviewer 2 Report
I think the manuscript was improved.
Author Response
Thank you for your review, we are uploading corrected manuscript (edited minor errors and English language)
